# Multi-Level Attention Split Network: A Novel Malaria Cell Detection Algorithm

Zhao Xiong * and Jiang Wu

School of Information Science and Engineering, Zhejiang Sci-Tech University,
Hangzhou 310018, China; wujiang@zstu.edu.cn
* Correspondence: 202130504186@mails.zstu.edu.cn

**Abstract:** Malaria is one of the major global health threats. Microscopic examination has been designated as the "gold standard" for malaria detection by the World Health Organization. However, it heavily relies on the experience of doctors, resulting in long diagnosis time, low efficiency, and a high risk of missed or misdiagnosed cases. To alleviate the pressure on healthcare workers and achieve automated malaria detection, numerous target detection models have been applied to the blood smear examination for malaria cells. This paper introduces the multi-level attention split network (MAS-Net) that improves the overall detection performance by addressing the issues of information loss for small targets and mismatch between the detection receptive field and target size. Therefore, we propose the split contextual attention structure (SPCot), which fully utilizes contextual information and avoids excessive channel compression operations, reducing information loss and improving the overall detection performance of malaria cells. In the shallow detection layer, we introduce the multi-scale receptive field detection head (MRFH), which better matches targets of different scales and provides a better detection receptive field, thus enhancing the performance of malaria cell detection. On the NLM—Malaria Dataset provided by the National Institutes of Health, the improved model achieves an average accuracy of 75.9% in the public dataset of Plasmodium vivax (malaria)-infected human blood smear. Considering the practical application of the model, we introduce the Performance-aware Approximation of Global Channel Pruning (PAGCP) to compress the model size while sacrificing a small amount of accuracy. Compared to other state-of-the-art (SOTA) methods, the proposed MAS-Net achieves competitive results.

**Keywords:** deep learning; YOLOv5; malaria detection; self-attention mechanism; detection head

## 1. Introduction

Malaria, as a prominent global public health issue, poses a significant threat to human well-being. Despite notable progress in malaria prevention since 2000, the annual incidence of over 200 million new cases remains alarmingly high, leading to a persistently elevated death toll. The malaria parasites that infect humans include Plasmodium falciparum, Plasmodium vivax, Plasmodium ovale, and Plasmodium malariae [1]. Among these, P. falciparum presents the most severe risk to human health. The life cycle of malaria parasites is intricate, encompassing various stages such as the ring stage, trophozoite stage, schizont stage, and gametocyte stage. Microscopic examination of Giemsa-stained blood smears continues to serve as the preferred method for definitively diagnosing both malaria parasites and malaria cases [2]. Nonetheless, the dependence on experienced physicians for conducting microscopic examination of blood smears remains unchanged. Following successful endeavors in malaria prevention and control, there has been a decline in the incidence of malaria cases. As a result, numerous primary healthcare facilities and infectious disease control institutions have redirected their focus away from conducting microscopic examinations for malaria. Consequently, there has been an enduring deterioration in the proficiency of malaria microscopy [3–5], resulting in an insufficient capacity to promptly identify cases of malaria [6].

The field of computer vision has experienced significant advancements in recent years, with deep learning-based neural network models achieving remarkable success in object detection. These advanced models have been widely applied in the field of biological image analysis. Utilizing computer-aided diagnosis can substantially alleviate the workload of healthcare professionals by facilitating the automatic identification of cells and their respective disease types. The proposed method effectively reduces false negatives and false positives resulting from visual fatigue, thereby providing fast and accurate quantitative results. These findings have significant implications for the prevention and control of malaria. EfficientNet, GoogleNet, and VGG19 are mainly used for the recognition and localization of malaria cells in malaria detection. If accurate identification of the types of malaria and judgment of their developmental stages are required, the Faster-RCNN and YOLO series algorithms are more commonly employed. In the past year, the RT-DETR method has been applied in related fields. However, the microscopic examination of malaria blood smears poses several challenges. Malaria parasites reside within red blood cells and undergo minimal changes in shape during different developmental stages. As a result, differentiation can only be achieved by observing the texture of the cytoplasmic staining. Furthermore, individual cells occupy a minuscule portion of the image, and a single blood smear typically contains hundreds of cells, with cells often adhering to and obstructing one another. These factors pose challenges for feature extraction. To address these challenges, it becomes essential to enhance the model's feature extraction capabilities and effectively utilize information. Hence, we propose MAS-Net, a context-aware feature extraction model. Computer-aided diagnosis can greatly alleviate the workload of healthcare professionals and facilitate the automatic identification of cells and their corresponding disease types. Additionally, it effectively reduces false negatives and false positives caused by visual fatigue while providing rapid and accurate quantitative results. This holds immense significance for malaria prevention and control. The specific approach is described as follows:

1. We introduce the Split Contextual Attention Structure (SPCot) module to incorporate contextual information into our feature extraction network. This module enhances the capability of feature extraction and also reduces the complexity of the model via the introduction of a redundant split convolutional (RSConv).
2. An MRFH detection head is proposed, introducing the adaptive matching receptive field to resolve the mismatch issue between the receptive field and the target size.
3. By using Performance-aware Approximation of Global Channel Pruning (PAGCP) [7] for pruning our model, we significantly reduces the parameter count of our model at the cost of sacrificing minimal accuracy.

## 2. Related Work

With the development of image processing and computer science technology, the detection of malaria-infected cells has gradually shifted from purely manual to semi-manual or even fully automated. A large number of computer vision techniques have been applied during the analysis of medical data. F. Boray Tek et al. [8] studied the self-detection and identification method of malaria parasites in thin blood smear images and found that Giemsa staining could highlight malaria parasites, white blood cells, platelets, and artificially introduced noise. Therefore, an improved K-nearest neighbor classifier-based binary parasite detection method was proposed. At this stage, malaria detection has transitioned from purely manual to semi-manual, while cell detection is still limited to manually designed features. In 2015, LeCun et al. [9] first proposed the theory of deep learning, which has great significance in the development history of neural networks. In recent years, deep convolutional neural networks have achieved good results in various medical image analysis and processing tasks, and now various machine learning and deep learning algorithms have been applied to the detection of malaria parasites. In 2017, Liang et al. [10] proposed a new machine learning model based on Convolutional Neural Networks (CNNs) to classify uninfected and parasitic cells. Vijayalakshmi [11] applied the Support Vector Machine (SVM) and VGG architecture for malaria detection. In [12–14],

pre-trained CNNs such as LeNet, AlexNet, GoogleNet, ResNet, and VGGnet were used with transfer learning techniques to detect malaria parasites, achieving high detection accuracy. Hung et al. [15] introduced a malaria detection method based on Faster R-CNN, which employed an object detection model previously used for natural images to identify malaria parasites in blood smear images. M. K. Dath [16] and S. S [17] used VGG19 for malaria detection and compared it with Restnet and SVM, proving the effectiveness of VGG19. Y. Jusman et al. [18] applied VGG19 and GoogLeNet for malaria parasite classification, and demonstrated that GoogLeNet is more suitable for the classification task compared to VGG19. F. A. S. Araujo et al. [19] employed EfficientNet for malaria cell detection and achieved good results. F. Yang [20] proposed the use of Cascade YOLO and YOLOv2. In [21], F. Abdurahman introduced the improved YOLOv3 and YOLOv4, which have higher accuracy and lower computational requirements. Due to efficiency issues with two-stage detection models, research on one-stage paradigm-based detection models has become increasingly popular. D. D. Acula et al. [22] compared the performance of DackNet53 and VGG16 models in malaria detection, and DackNet53 achieved good results. Liu Z et al. [23] used YOLOv5 for detecting malaria cells in blood smears and achieved excellent results. Guemas E et al. [24] introduced RT_DETR for malaria detection, which has lower accuracy compared to mainstream models, but has extremely low parameters and computational complexity. In recent years, both CNN and Transformer have made new research achievements. J. Li proposed SCConv [25], a plug-and-play module with less system redundancy. Y. Li et al. proposed a new self-attention structure called Transform, enhancing the feature extraction capability for small targets [26]. Li C et al. proposed ODConv [27], which is based on multi-channel attention and dynamic convolution. Regarding feature extraction networks, S.-H. Gao introduced Res2Net at a granularity level, representing multi-scale features and increasing receptive fields for each network layer [28]. S. Woo et al. proposed ConvNeXt V2, which approximates the performance of Transformer models [29]. Yu W. et al. introduced InceptionNeXt based on large convolution kernels, and there have also been lightweight works such as GhostNet V2, FasterNet, and EdgeNeXt [30–33]. Transformer-based models, such as SwinTransformer, EfficientViT, and RepVit [34–36], have shown good performance in complex scenes and small object detection. Among them, YOLOv5, proposed in 2021, has achieved a good balance between accuracy and efficiency, surpassing most of the two-stage detection models known for high accuracy. Thus, it can serve as a benchmark for current detection models.

## 3. MAS-Net

In this section, we will provide a detailed introduction to the proposed MAS-Net. The overall network architecture is shown in Figure 1.

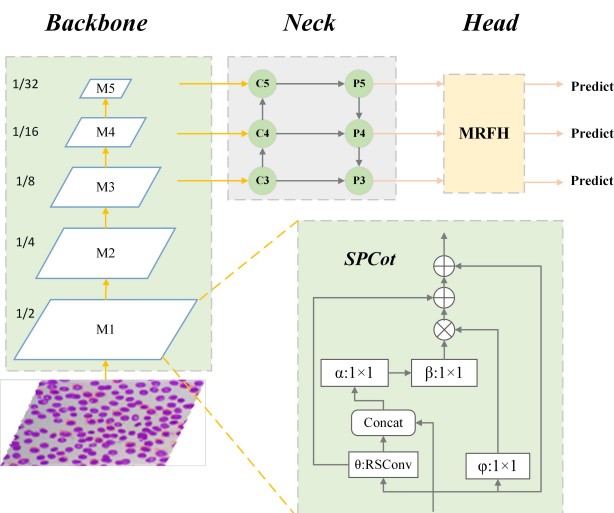

**Figure 1.** Network structure of MAS-Net. Following the general paradigm, the overall architecture of MAS-Net consists of three components: backbone, neck, and head.

MAS-Net follows the general paradigm of backbone, neck, and head. Firstly, SPCot is used as the unit to construct the feature extraction network, which outputs features maps of three scales: 1/8, 1/16, and 1/32. The specific backbone structure is shown in Figure 2. For the neck, we adopt the Feature Pyramid Network (FPN) structure from YOLOv5. Finally, the features are sent into the MRFH detection head for optimal receptive field adaptation.

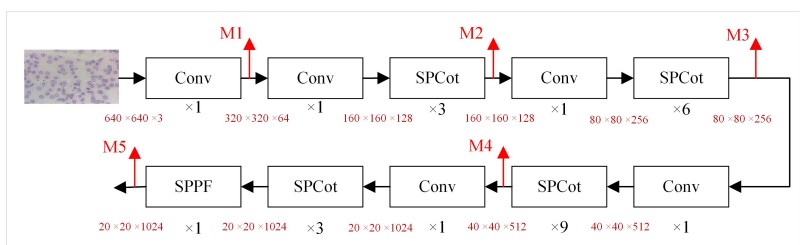

**Figure 2.** Backbone of MAS-Net. The "SPPF" in the diagram is from the YOLOv5 network. The outputs of layers 5, 7, and 10 are represented as M3, M4, and M5.

### 3.1. RSConv

To minimize information loss and facilitate feature extraction in multiple channels, our model avoids extensive channel reduction and expansion operations. However, this approach leads to an exponential growth of parameters. To improve the efficiency of our model, we have explored techniques such as MobileNets [37–39], ShuffleNets [40,41], and Ghost-Net [42]. These methods utilize depthwise convolution (DWConv) [43] and group convolution (GConv) [44] to extract spatial features. While this approach helps reduce the number of floating-point operations (FLOPs), it does not effectively address the issue of reducing memory access [32]. Previous studies [32,45] have suggested minimizing redundancy in the feature maps. Inspired by this observation, we designed the segmentation-based RSConv method, illustrated in Figure 3.

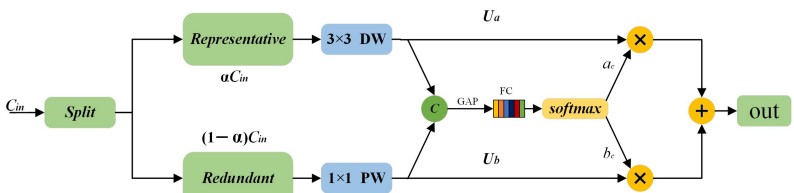

**Figure 3.** RSConv structure diagram, where "Split" denotes channel splitting operation, "GAP" represents global average pooling, and "FC" stands for fully connected layer.

The input feature map is denoted as $X \in R^{c \times h \times w}$ and the output feature is $Y \in R^{m \times h \times w}$, with $c$ representing the input channel and m representing the output channel. We define $W \in R^{c \times k \times k \times m}$ as the convolution weights with a kernel size of $k \times k$. The mathematical expression of the convolution operation, given by $Y = WX + b$, can be expressed as follows:

$$
\begin{bmatrix} y_1 \\ y_2 \\ \vdots \\ y_m \end{bmatrix} = \begin{bmatrix} W_{11} & W_{12} & \dots & W_{1,c} \\ W_{21} & W_{12} & \dots & W_{2,c} \\ \vdots & \vdots & \ddots & \vdots \\ W_{m,1} & W_{m,2} & \dots & W_{m,c} \end{bmatrix} \begin{bmatrix} x_1 \\ x_2 \\ \vdots \\ x_c \end{bmatrix}
\tag{1}
$$

$x_i, i \in [1, c]$ represents one of the channels of the input feature map, and $x_i, i, j \in [1; c, m]$ represents one of the m convolution kernels. After the convolution, we obtain the output $y_j, j \in [1, m]$. RSConv divides the input channels into two main components: one component applies a $k \times k$ depth-wise convolution (DWConv) to capture intrinsic information, and the other component redundantly applies a cost-effective $1 \times 1$ convolution to enhance

subtle hidden details. The left side of Figure 3 illustrates this configuration. Therefore, our initial RSConv can be represented as:

$$
\begin{bmatrix} y_1 \\ y_2 \\ \vdots \\ y_m \end{bmatrix} = \begin{bmatrix} W_{11} & W_{12} & \dots & W_{1,\alpha c} \\ W_{21} & W_{12} & \dots & W_{2,\alpha c} \\ \vdots & \vdots & \ddots & \vdots \\ W_{m,1} & W_{m,2} & \dots & W_{m,\alpha c} \end{bmatrix} \begin{bmatrix} x_1 \\ x_2 \\ \vdots \\ x_{\alpha c} \end{bmatrix} + \begin{bmatrix} W_{1,\alpha c+1} & W_{1,\alpha c+2} & \dots & W_{1c} \\ W_{2,\alpha c+1} & W_{1,\alpha c+2} & \dots & W_{2c} \\ \vdots & \vdots & \ddots & \vdots \\ W_{m,\alpha c+1} & W_{m,\alpha c+2} & \dots & W_{mc} \end{bmatrix} \begin{bmatrix} x_{\alpha c+1} \\ x_{\alpha c+2} \\ \vdots \\ x_c \end{bmatrix} \tag{2}
$$

where $W_{ij}, j \in [1, \alpha c]$, represents the parameters of a $3 \times 3$ kernel for convolution on channel $\alpha c$. $W_{ij}, j \in [\alpha c + 1, c]$, represents the parameters of the inexpensive $1 \times 1$ kernel, which operates through point convolution on the remaining $(1 - \alpha)c$ redundant features.

Combining feature maps with different receptive fields directly may lead to the loss of crucial geometric details and introduce conflicts in information. In the end, we choose to connect the feature maps from different branches along the channel to preserve both spatial and channel information in the resulting feature maps. Consequently, the preserved feature maps can be utilized to compute the relative importance of each channel. We obtain channel statistics data $S_c$ by applying global average pooling, thus incorporating global information.

$$
S_c = F_{gap(U_c)} = \frac{1}{H \times W} \sum_{i=1}^{H} \sum_{j=1}^{W} U_c(i, j) \tag{3}
$$

Using a fully connected layer ($f_c$) to obtain feature $z$, which guides precise and adaptive selection:

$$
z = F_{fc}(S_c) \tag{4}
$$

We combine these two results, $z_a$ and $z_b$, by stacking their vectors together, and then perform cross-channel soft attention operations. $a_c$ and $b_c$ represent the soft attention vectors of $U_a$ and $U_b$.

$$
a_c = \frac{e^{z_a}}{e^{z_a} + e^{z_b}}, \quad b_c = 1 - a_c, \tag{5}
$$

The final output $Y$ can be obtained by fusing features $U_a$ and $U_b$:

$$
Y = a_c \cdot U_a + b_c \cdot U_b, \quad a_c + b_c = 1 \tag{6}
$$

Complexity analysis:
The parameters of a regular convolution can be calculated as:

$$
P_{Conv} = k \times k \times c \times m \tag{7}
$$

The parameters of RSConv can be calculated as:

$$
P_{RSConv} = k \times k \times \alpha c + 1 \times 1 \times \alpha c \times m + 1 \times 1 \times (1 - \alpha)c \times m \tag{8}
$$

After simplification, we obtain:

$$
P_{RSConv} = k \times k \times \alpha c + c \times m \tag{9}
$$

When we set $\alpha = 0.5$ and use a $3 \times 3$ kernel, the parameters can be reduced to $1/9$ of the original.

### 3.2. Split Contextual Attention Structure

The YOLOv5 baseline network employs Darknet53 for extracting features. Malaria cells are found in blood smears in a small proportion and frequently overlap and adhere to one another. These cells are mainly characterized by subtle texture features. Additionally, neighboring cells create significant interference. Furthermore, scaling or reducing the

channel of the feature map inevitably results in the loss of information about smaller targets. Drawing inspiration from [26,46], we introduce the self-attention structure, SPCot, which integrates contextual information. The structure is shown in Figure 4.

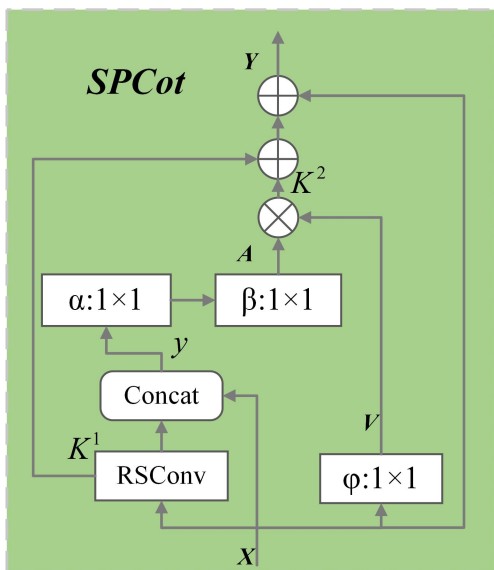

**Figure 4.** SPCot structure diagram. SPCot enhances detection performance by combining local and global attention. $\alpha$, $\beta$, $\varphi$ represent $1 \times 1$ convolutions, while RSConv is used to extract features and reduce the number of parameters. SPCot builds the backbone as a feature extraction module, which you can find in Figure 2.

SPCot utilizes RSConv to encode the contextual information of all adjacent keys within a spatially oriented $3 \times 3$ grid, resulting in the contextual information $K^1 \in R^{H \times W \times C}$. Subsequently, the local static contextual information $K^1$ is concatenated with the original input feature map $X$. Two consecutive 1 convolutions are then used to obtain the attention matrix $A$. In this case, $X$ represents the original input feature map, while $W_\alpha$ and $W_\beta$ represent the two convolution layers. The calculation formula is expressed as follows:

$$A = [K^1, X] W_\alpha W_\beta \tag{10}$$

In this context, $X$ represents the input feature information, while $V$ is defined as $V = XW_\varphi$, where $W_\varphi$ represents a $1 \times 1$ convolution weight matrix. Afterwards, a matrix multiplication is performed to obtain the global dynamic contextual information $K^2$. The calculation formula is given as follows:

$$K^2 = V \cdot A \tag{11}$$

By fusing the local static contextual information $K^1$ and the global dynamic contextual information $K^2$, the output is obtained.

$$K = K^1 + K^2 \tag{12}$$

To mitigate potential information loss resulting from convolutions, residual structures are employed to compensate for the features.

$$Y = K + X \tag{13}$$

The detailed process can be described as Algorithm 1.

---

**Algorithm 1** Algorithm for self-attention module based on contextual information

---

**Input:** $X$ [batch, channel, height, weight]
**Output:** The final output $Y$

1: $K^1 = RSConv(X)$. The feature map $X$ employs the RSConv convolution operation to acquire the local static contextual information $K^1$.
2: $V = XW_\varphi$. The feature map $X$ undergoes a $1 \times 1$ convolution process to derive the output $V$.
3: $A = [K^1, X]W_\alpha W_\beta$. $A$ is the attention matrixn.
4: $K^2 = V \cdot A$. The global dynamic contextual information $K^2$ is obtained through element-wise multiplication of $V$ and the attention matrix$A$.
5: $K = K^1 + K^2$. The global contextual information $K$ is obtained by adding $K^1$ and $K^2$.
6: $Y = X + K$. The final output $Y$ is obtained by adding the global contextual information $K$ to the input features $X$.

---

We utilize the SPCot structure to construct our feature extraction network. The main construction parameters of the backbone are presented in the Table 1. The detailed network structure is shown in Figure 3.

**Table 1.** The backbone network for feature extraction uses the following parameters.

| Output Channel | Convolution Size | Number of Repetitions |
|---|---|---|
| 64 | Conv(kernel:$3 \times 3$,stride 2) | 1 |
| 128 | Conv(kernel:$3 \times 3$,stride 2) | 1 |
| 128 | SPCot | 3 |
| 256 | Conv(kernel:$3 \times 3$,stride 2) | 1 |
| 256 | SPCot | 6 |
| 512 | Conv(kernel:$3 \times 3$,stride 2) | 1 |
| 512 | SPCot | 9 |
| 1024 | Conv(kernel:$3 \times 3$,stride 2) | 1 |
| 1024 | SPCot | 3 |
| 1024 | SPPF | 1 |

*3.3. Multi-Scale Receptive Field Detection Head*

The detection head in the Dynamic Scale-Aware Head YOLO series typically comprises a $3 \times 3$ convolutional layer followed by a $1 \times 1$ convolutional layer. Due to the specific nature of the detection targets, a single type of detection head may exhibit a discrepancy between the receptive field and the target size. When the receptive field is too small, it can only capture local features, rendering it insufficient to capture comprehensive information about the target. Conversely, an excessively large receptive field introduces excessive noise and irrelevant information, leading to suboptimal detector performance. In their study, ref. [47] presented a dynamic selection mechanism in CNN that enables each neuron to adaptively adjust its receptive field size based on the multiple scales of input information. Leveraging this mechanism, we developed the MRFH detection head, as depicted in Figure 5.

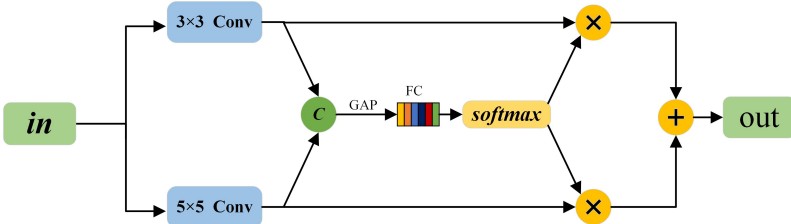

**Figure 5.** MRFH structure diagram. the detection head MRFH utilizes a $5 \times 5$ convolution achieved by a $3 \times 3$ convolution with a dilation rate of 2.

After passing the input into MRFH, a series of convolutions are applied to the feature map. Specifically, a $3 \times 3$ convolution and a $5 \times 5$ convolution (achieved by two $3 \times 3$ convolutions with a dilation rate of 2) are used. These convolutions utilize diverse kernels to establish distinct receptive fields. Subsequently, softmax attention is employed to combine the information from these branches, allowing for the fusion of branches with different receptive fields. It is well known that simply summing feature maps with varying receptive fields can result in the dominance of specific geometric details and introduce conflicting information. Additionally, the reduction in channel of the obtained feature maps hampers their ability to effectively guide the learning process of both branches simultaneously. In contrast to SKNet's [47] approach of pixel-wise summation followed by average pooling, we adopt an alternative strategy. Firstly, separate pooling operations are performed on each branch, and then their results are concatenated instead of using pixel-wise addition. We have decided to connect the feature maps from different branches in the channel in order to obtain feature maps that preserve both spatial and channel information. This enables the computation of the relative importance of each channel. This approach not only allows for the independent determination of channel importance for a specific receptive field, but also identifies the importance of feature maps relative to other branches with distinct receptive fields. In a previous study [48], it was demonstrated that a $3 \times 3$ convolutional layer fails to provide adequate receptive fields for each branch, leading to a preference for branches with larger receptive fields. Shallow detection heads, in particular, necessitate larger receptive fields to incorporate more global contextual information, thereby enhancing the recognition of small targets. Meanwhile, deep detection heads have already achieved significant receptive fields through the stacking of multiple residual modules and the introduction of SPPF modules. However, integrating MRFH into the deep detection heads quickly reaches a saturation point and introduces a higher number of parameters, resulting in inefficiency. Taking into consideration the balance between detection accuracy and model efficiency, we have made the decision to exclusively employ the MRFH detection head in the shallow detection layers.

*3.4. PAGCP*

Taking into consideration the limited computing resources commonly available in malaria-endemic areas, we employed the PAGCP [7] method to prune and compress the model, resulting in a more suitable MAS-Net-Tiny model for practical deployment. PAGCP adjusts the pruning optimization objective to:

$$\min_{\theta \in \Omega} \mathbb{E}_{x \sim D_x}[\mathcal{S}(x; \theta)] \quad \text{s.t. } \mathrm{g}(\theta) \leq \alpha \tag{14}$$

$\mathcal{S}(\cdot)$ represents the significance standard used to evaluate the importance of filters. $x$ represents the input image, where $x$ follows the distribution of $D$. Additionally, $\mathrm{g}(\cdot)$ represents a constraint vector determined by several factors, such as the reduction rate of FLOPs and parameters, while $\alpha$ is a threshold vector in which each channel corresponds to each of the aforementioned factors. $\Omega$ is the set of all pruning selections in the original model. $l = 1, 2, \ldots, L$ and $k = 1, 2, 3, \ldots, K_l$, where $K$ is the number of filters in the $l$-th layer. $\Omega$ represents the total set of parameters. The constraint function $\mathrm{g}(\cdot)$ is parameterized by $\theta$, where $\theta = 0$ indicates the deletion of the convolutional kernel, while $\theta = 1$ preserves it. The performance constraint is designed as:

$$\mathrm{g}(\theta) = \|\Delta \mathcal{L}_1(\theta), \Delta \mathcal{L}_2(\theta), \ldots, \Delta \mathcal{L}_T(\theta)\|_\infty \tag{15}$$

$$= \max_{1 \leq t \leq T} |\Delta \mathcal{L}_t| \tag{16}$$

The layer-wise constrained boundary is designed in a cascading transformation form:

$$\prod_{i=1}^{L} \left(1 + d_1 \lambda^{i-1}\right) = \alpha \tag{17}$$

Among them, $d_1$ represents the constraint boundary during the initial layer pruning, which is the value of $g(\theta)$. $\lambda$ is the constraint boundary for each optimization step, equivalent to the scaling factor of the constraint boundary in the previous step. $\alpha$ is the global constraint boundary.

PAGCP adjusts the optimization objective to perform a constrained minimization search only on saliency indicators while keeping the model parameter weights unchanged. Among them, the constraint terms are adjusted to include performance loss constraints after model pruning and computational constraints. This ensures that the model parameters are in the optimal state and enables more accurate importance evaluation.

## 4. Model Training and Result Analysis

### 4.1. Dataset Introduction

We employed the Plasmodium vivax (malaria) infected human blood smear dataset [49] in this study. It consists of about 1364 images in PNG format. The dataset includes approximately 80,000 cells stained with the Giemsa stain. It covers two categories of uninfected cells: red blood cells and white blood cells. Moreover, it consists of four categories of infected cells: gametocytes, schizonts, merozoites, and trophozoites. The dataset's annotation covers all these classes, including challenging-to-identify cells. Challenging cells are labeled as difficult types. Each image has a resolution of $1600 \times 1200$ pixels. In the experiment, we excluded cells labeled as difficult, white blood cells, and uninfected red blood cells. We retained only infected cells for further analysis.

### 4.2. Experimental Environment

The experimental operating system of this article is Windows 10, completed based on the GPU, PyTorch, and CUDA framework. The CPU of the experimental environment is an AMD Ryzen 7 5800, the graphics card is an NVIDIA RTX 3090 with 24 GB of memory, and the operating memory is 16 GB. Under the above experimental environment, the image dataset is being trained. The learning rate decay strategy is based on the cosine learning rate decay [50], with SGD chosen as the optimizer, and momentum and weight decay values are set to 0.937 and 0.0005, respectively. The batch size is set to 16, and no pre-trained weights are being loaded. The training consists of 300 epochs. The image augmentation strategies include Mixup, Mosaic, HSV transformation, and affine transformation, as well as image concatenation. The loss function combines CIOU and applies NWD [51]. In the PAGCP pruning process, an initial rate of 0.05 is set, an initial threshold of 5 is set, and other parameters are set to default values. It runs for 100 epochs.

### 4.3. Results and Analysis

This experiment evaluates precision (*Pre*), recall rate (*Rec*), mean Average Precision (*mAP*), floating-point operations (*FLOPs*), $F_1$ score, and frames per second (*FPS*).

### 4.4. Grad-CAM

Figure 6 shows the Grad-CAM of MAS-Net. In Figure 6a, it is difficult for our naked eye to detect the infected target at the center of the image, while our network can accurately focus on it. Through Figure 6b, we can clearly see the region of interest that the network is attentive to, which corresponds to the distribution of heme pigment. Additionally, we can observe that the network also pays attention to the swelling of infected cells, which is consistent with the characteristics of manually inspecting malaria cells.

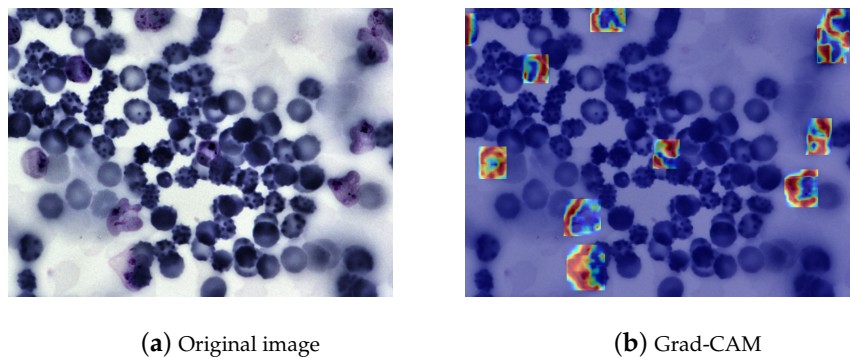

(**a**) Original image          (**b**) Grad-CAM

**Figure 6.** Grad-CAM of MAS-Net. We output the heat map of the shallow layers of the network, where the red part represents the network's region of interest. The deeper the color, the more interested the network is in that area..

### 4.5. Experimental Results

We compared the MAS-Net model with other YOLO series models. The results showed that compared to other one-stage models, MAS-Net performed better in terms of the PR curve, as shown in Figure 7. Additionally, there was a significant improvement in mAP value, as detailed in Figure 8.

According to Figure 7, it is evident that our MAS-Net model exhibits a superior precision–recall curve. Furthermore, Figure 8 highlights a slight reduction in the convergence speed of MAS-Net compared to other models. This can be attributed to the heightened depth caused by the complex structure of SPCot, which subsequently affects the training speed. Notably, MAS-Net achieves a remarkable $mAP$ of 75.9% at 280 epochs, surpassing the performance of other SOTA models.

We can see from Table 2 that MAS-Net outperforms other models in terms of precision, $recall$, $mAP$, and $F_1$ score. Although MAS-Net has higher parameter and computational complexity compared to other models, it is lower than the v8 model with similar accuracy. However, MAS-Net has a slower image processing speed, which limits its practical applicability, especially in resource-constrained environments. To address this issue, we compressed the MAS-Net model and obtained MAS-Net-Tiny through PAGCP channel pruning.

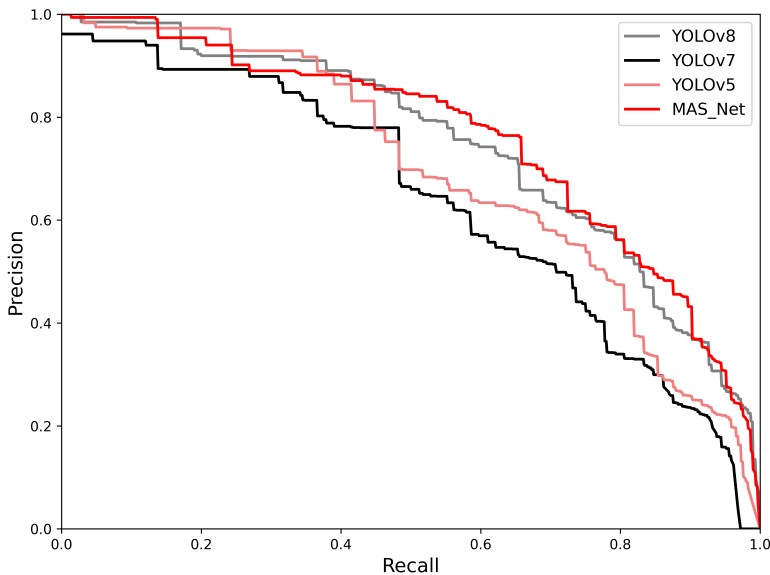

**Figure 7.** The *PR* curve graph. It shows the *PR* curve of mainstream one-stage models.

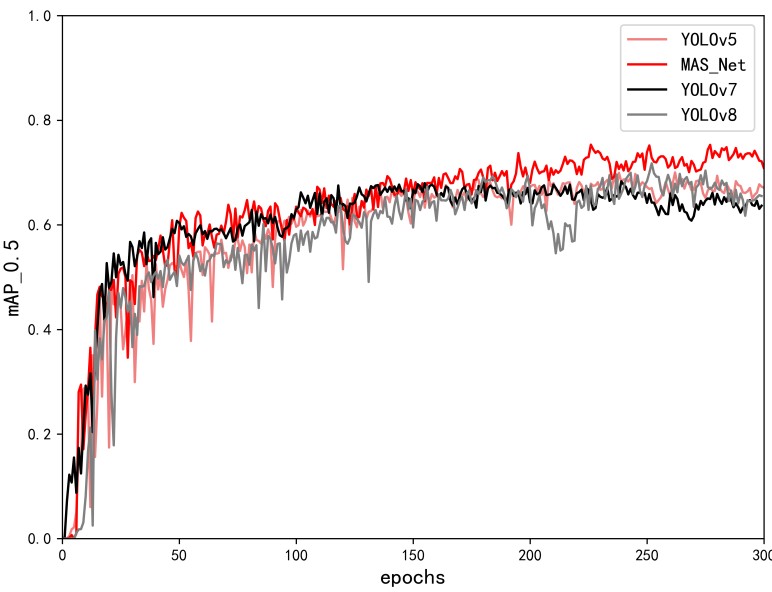

**Figure 8.** The *mAP* variation curve. The figure shows the *mAP* curve of the main SOTA models on our dataset.

**Table 2.** Comparison of MAS-Net with other mainstream one-stage detection algorithms.

| Algorithm | *Pre*% | *Rec*% | *mAP*% | *Para* | *GFLOPs* | $F_1$ | *FPS* |
|---|---|---|---|---|---|---|---|
| YOLOv5 | 63.4 | 67.8 | 69.9 | 46.3 m | 108.3 | 65.5 | 46 |
| YOLOv7 | 62.9 | 66.9 | 67.8 | 39.2 m | 105.2 | 64.8 | 50 |
| YOLOv8 | 68.8 | 65.3 | 71.7 | 43.6 m | 164.8 | 66.0 | 62 |
| MAS-Net | 73.9 | 72.2 | 75.9 | 48.7 m | 126.2 | 73.0 | 42 |
| MAS-Net-Tiny | 62.6 | 69.8 | 70.6 | 9.6 m | 36.8 | 66.0 | 125 |

From Table 3, it can be seen that our model has made significant progress in simplifying complexity and has also improved in processing speed. This helps improve the applicability of the algorithm in practical applications, but it has also led to a decrease of 5.3% in mAP.

**Table 3.** Parameter comparison before and after PAGCP.

| Algorithm | *mAP*% | *Para*% | *GFLOPs*% | *FPS* | *mAP* ↓ | *Para* ↓ (%) | *GFLOPs* ↓ (%) |
|---|---|---|---|---|---|---|---|
| MAS-Net | 75.9 | 48.7 m | 126.2 | 42 | - | - | - |
| MAS-Net-Tiny | 70.6 | 9.6 m | 36.8 | 125 | 5.3 | 80.2 | 70.7 |

The downward arrow (↓) indicates a decrease.

Compared to the EfficientNet method in [19], MAS-Net has a higher computational complexity but slight advantages in mAP. Additionally, MAS-Net also has a faster image processing speed. Our models have shown improvements in performance compared to the models introduced in [23], which incorporate CA and BiFPN. The RT-DETR model in [24] has the advantage of low computational complexity and a high detection speed, as shown in the table. By comparison, our MAS-Net-Tiny model achieves a higher mAP and FPS with lower parameters and computational complexity.

To validate the effectiveness of our model, we compared it with other detection networks. The results are shown in Table 4.

**Table 4.** Comparison of MAS-Net with other mainstream one-stage detection algorithms.

| Algorithm | *Pre%* | *Rec%* | *mAP%* | *Para* | *GFLOPs* | *F₁* | *FPS* |
|---|---|---|---|---|---|---|---|
| Efficientnet [19] | 62.3 | 74.5 | 71.8 | 49.2 m | 80.8 | 67.9 | 10 |
| EfficientnetV2 [52] | 64.7 | 65.1 | 72.2 | 74.1 m | 134.0 | 64.9 | 13 |
| ConvnextV2 [29] | 61.3 | 60.3 | 64.3 | 50.6 m | 113.2 | 59.8 | 29 |
| MobilenetV3 [37] | 57.3 | 61.3 | 64.6 | 23.2 m | 43.1 | 59.2 | 28 |
| Fasternet [32] | 63.8 | 65.7 | 68.1 | 23.7 m | 44.4 | 59.2 | 28 |
| ResnetV2 [53] | 61.5 | 65.0 | 64.8 | 51.2 m | 113.0 | 63.2 | 39 |
| Edgenext [33] | 58.8 | 64.8 | 63.7 | 40.1 m | 87.6 | 61.7 | 30 |
| EfficientViT [35] | 68.7 | 63.3 | 68.5 | 33.8 m | 68.0 | 65.9 | 17 |
| RT-DETR [54] | 67.9 | 65.4 | 61.9 | 32.0 m | 87.2 | 66.6 | 100 |
| MAS-Net | 73.9 | 72.2 | 75.9 | 48.7 m | 126.2 | 73.0 | 42 |
| MAS-Net-Tiny | 62.6 | 69.8 | 70.6 | 9.6 m | 36.8 | 66.0 | 125 |

We compared our model with other SOTA models using the PAN structure and trained it for 300 epochs. The results are shown in Table 5. Compared to other models, MAS-Net has higher accuracy but also higher complexity. On the other hand, MAS-Net-Tiny achieves a good balance between computational complexity and accuracy, which is advantageous for the model's practical applicability.

According to the results presented in Table 2, the accuracy of detecting malaria cells with MAS-Net reaches 75.9%, which demonstrates a significant 6.9% improvement in $mAP$ compared to the baseline YOLOv5 model. Moreover, precision has seen a notable increase of 10.5%, whereas recall has shown a substantial improvement of 4.4%. Furthermore, the computational complexity, measured in terms of $GFLOPs$, has increased by 15.7%. Moreover, considering the findings outlined in Table 5, it becomes clear that the increase in parameters can be primarily attributed to our MRFH detection head, whereas the deeper layers of MRFH contribute marginally to the improvement in $mAP$. Therefore, we chose not to employ MRFH in the deeper layers. After adopting PAGCP, the model's mAP value decreased by 5.3%. However, there was a significant simplification in terms of parameters and computational complexity. The experimental results demonstrate that the MAS-Net model achieved high recognition accuracy in the detection of malaria-infected cells after Giemsa staining. Furthermore, the model's parameter rationality and performance are superior to other state-of-the-art (SOTA) models.

**Table 5.** Ablation experiments. Baseline network is YOLOv5, and all experiments are conducted using mosaic data augmentation method.

| Mosaic | SPCotNet | MRFH | PAGCP | *mAP%* | *GFLOPs* | Para (Million) |
|---|---|---|---|---|---|---|
| √ | - | - | - | 69.0 | 108.3 | 47.9 |
| √ | √ | - | - | 75.5 | 116.2 | 46.6 |
| √ | √ | √ | - | 75.9 | 126.2 | 48.7 |
| √ | √ | * | - | 75.3 | 156.6 | 90.6 |
| √ | √ | √ | √ | 70.6 | 36.8 | 9.6 |

The asterisk (∗) indicates the use of MRFH detection head in all detection layers. The checkmark (√) represents adopting the structure, while the dash (-) represents not adopting the structure. Using MRFHdetection heads in deep neural networks comes at a high cost, so MAS-Net only employs MRFH detection heads in shallow neural networks.

Figure 9 illustrates the detection results of this model on the dataset, comprising six cell categories: four types of malaria-infected cells, healthy red blood cells, and white blood cells. The model exclusively detects the malaria-infected cells, yielding results that include predicted bounding boxes, class labels (ri: ring, tr: trophozoite, sc: schizont, ga: gametocyte), and confidence scores. Upon comparing the results, it becomes evident that the original YOLOv5 model exhibits a lack of confidence in accurately predicting the bounding boxes containing the targets, resulting in subpar detection performance with numerous missed instances. In contrast, our model effectively resolves the aforementioned

issues. The confidence scores are generally higher in comparison to the original model, resulting in fewer missed detections, particularly in densely populated regions where infected cells are small and cellular adhesion is prevalent. This improved performance is attributed to the SPCot structure, which extracts contextual information, enhancing the capabilities of the MAS-Net model compared to the original YOLOv5 model.

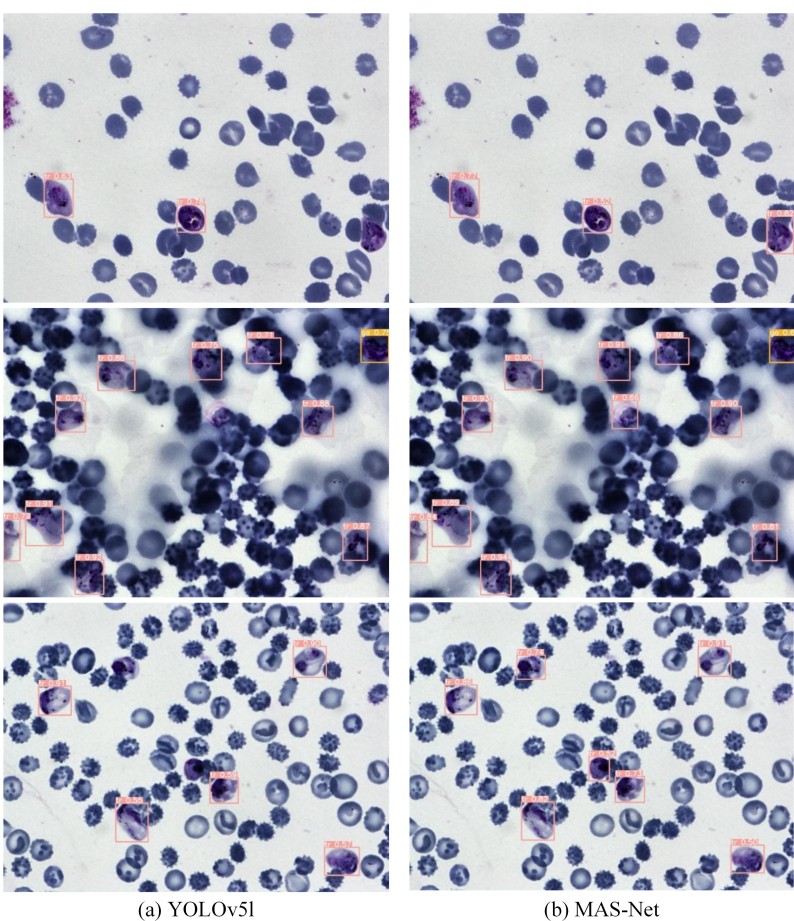

<div align="center">(a) YOLOv5l          (b) MAS-Net</div>

**Figure 9.** Detection results example.The figure shows a comparison of the detection performance between our model and the baseline model. The results include predicted bounding boxes, class names (ri: ring, tr: trophozoite, sc: schizont, ga: gametocyte), and confidence scores. MAS-Net can detect some hard-to-find targets, and it also achieves higher confidence scores.

The process of blood smear preparation and examination may result in low-quality images due to factors such as lighting conditions and microscope quality. We attempted to simulate this scenario by adding various types of noise and motion blur in the images. The detection results are shown in Figure 10.

Our model has a certain robustness to noise and can effectively identify infected cells. However, the confidence scores are relatively lower compared to the original images. On the other hand, our model exhibits high robustness to motion blur, with confidence scores similar to the original images. It may be because our model is more sensitive to colors in the images. It is difficult for the model to detect overlapping cells in the presence of noise, making it challenging to distinguish them as separate cells.

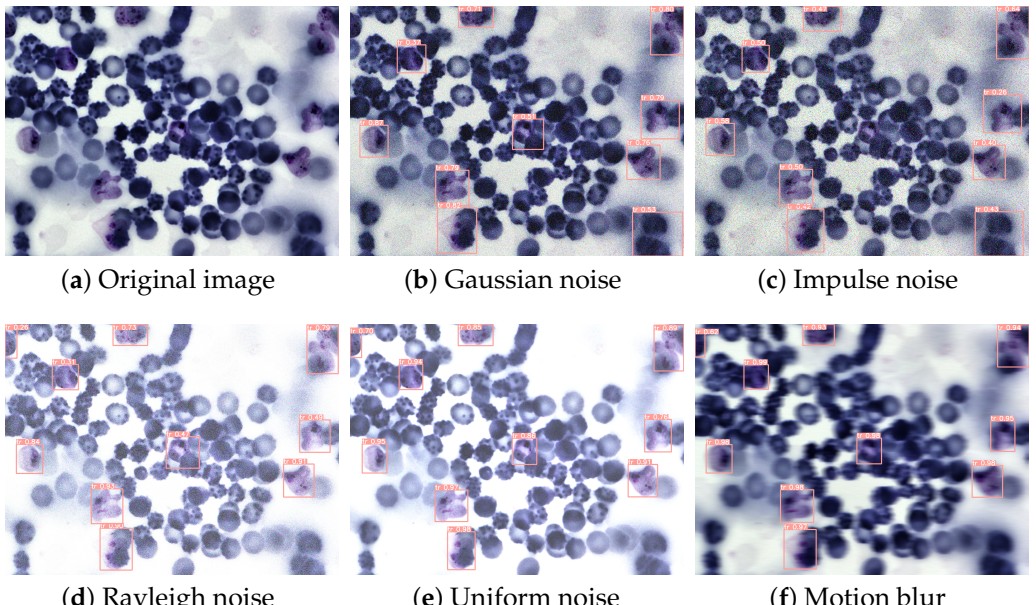

| (**a**) Original image | (**b**) Gaussian noise | (**c**) Impulse noise |
| (**d**) Rayleigh noise | (**e**) Uniform noise | (**f**) Motion blur |

**Figure 10.** Robustness test results visualization. Gaussian noise, salt-and-pepper noise, Rayleigh noise, uniform noise, and motion blur were added and tested accordingly. The test results consist of predicted bounding boxes, class names (ri: ring, tr: trophozoite, sc: schizont, ga: gametocyte), and confidence scores.

## 5. Conclusions

Applying computer science and technology to the field of biomedical research holds great potential. We have proposed the MAS-Net algorithm for the detection and recognition of malaria parasites in blood smear cells. Our SPCot structure effectively leverages contextual information to extract more detailed information. During the network construction process, we reduce the loss of feature information by minimizing channel compression operations and employing RSConv to reduce computational parameters, avoiding parameter explosion. The design of the MRFH detection head provides a larger receptive field for shallow networks. By introducing PAGCP channel pruning, we compress the model to improve its practical applicability. Experimental results have demonstrated the effectiveness and robustness of our model. Using the methods described in this paper, we can significantly improve the detection rate and reduce the workload of relevant personnel, even in the absence of specialized microscopic technicians. This has significant implications, particularly in medical diagnosis within the field of healthcare.

## 6. Future Work

The training dataset suffers from class imbalance, which is the reason why our model's detection performance is only 75.9%. In our future work, we will attempt to address this issue by expanding the dataset through unsupervised learning, aiming to improve the recognition ability of our network and validate its generalization on diverse datasets. We also plan to extend the detection scope to malignant malaria parasites and other species of malaria parasites, as well as other bloodborne diseases that require microscopic examination. Furthermore, we will actively seek cooperation to promote the practical application of our detection network in diagnosis.

**Author Contributions:** Conceptualization, Z.X. and J.W.; Methodology, J.W.; Software, Z.X.; Verification, Z.X.; Resources, J.W.; Data Management, J.W.; Writing—First draft preparation, Z.X.; Writing—Review and Edit, J.W.; Project Management, Z.X. All authors have read and agreed to the published version of the manuscript.

**Funding:** This research was supported by Zhejiang 14th five-year graduate education reform project under Grant No. syjsjg2023144.

**Institutional Review Board Statement:** Not applicable.

**Informed Consent Statement:** Not applicable.

**Data Availability Statement:** The dataset used in this study can be found at the link below https://lhncbc.nlm.nih.gov/LHC-research/LHC-projects/image-processing/malaria-datasheet.html (accessed on 17 January 2024). The MAS-Net code can be found at the link below https://github.com/glassxiong/MAS_NET (accessed on 17 January 2024).

**Conflicts of Interest:** The authors declare no conflicts of interest.

## Abbreviations

The following abbreviations are used in this manuscript:

| | |
|---|---|
| MAS-Net | multi-level attention split network |
| SPCot | split contextual attention structure |
| MRFH | multi-scale receptive field detection head |
| RSConv | redundant split convolutional |
| PAGCP | Performance-aware Approximation of Global Channel Pruning |
| NWD | Normalized Wasser-stein Distance |

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
