# Peer review of "Multi-Level Attention Split Network: A Novel Malaria Cell Detection Algorithm"

_information, doi:10.3390/info15030166_

Round 1

Reviewer 1 Report

Comments and Suggestions for Authors

The paper introduces a deep learning architecture tailored to the detection and classification of malaria cells. Overall the paper introduces some interesting modifications to the standard Yolo architecture. In most parts the paper is well written. Some of the major drawbacks are:

1.    The paper is strongly focused on technical aspects without any background in biology or medicine. Towards this end the paper is mainly related to the probleprm of “Small Object Detection”. However, this topic is not discussed in the paper. The related work section should mainly focus on this topic.

2.    The presented results are not totaly sound. The proposed model is compared against standard Yolo architectures, which I think is not an adequate competitor to the stated problem. There already exist an extensive literature for the detection of small objects which should be considered here. There already exist some work on malaria cell classification and/or detection which should also be used for comparison. The major step the paper takes is to apply a detection algorithm instead of marking individual cells and doing classification only. This should be emphasized stronger in the experimental section, as well as throughoutt the rest of the article.

Some detailed comments:

1.    The paper is not well structured. The introduction should give some general information about the application field, especially on existing technical approaches. This is currently mainly included in the “Related Work Section”. The related work should mainly address approaches to small object detection pipelines.

2.    Keywords and abbreviations should be explained when or before using them. Examples are l.208 SKNet, l.230 IOU, l.125 RSConv, and many more.

3.    Section 2.3 is not novel This metric has been already been used extensively for small object detection problems.

4.    Section 3.3. is standard and can be shortened

5.     The overall training method is no described. Is pre-training, augmentation used ?

6.    Figure 8 is hardly readable. Further, symbols in figures should be explained in the caption.

7.    Figure 1: The SPCot block can not be located in the architecture and it is not explained in the caption.

8.    Section 2.1. is unclear in many parts

9.    The term channel and dimensions should be used consistently with the existing literature (e.g. l.154)

10. The bibliography should be supplemented. There allready exist several papers on malaria cell detection and classification. Further, references to small object detections should be added.

Comments on the Quality of English Language

The paper is well written.

Reviewer 2 Report

Comments and Suggestions for Authors

This paper proposed a multi-level attention split network (MAS-Net), the split contextual attention structure (SPCot) , MRFH, CIOU, and NWD for the dtetction of malaria cells. The model achieves an average accuracy of 75.9 % on the publicly available NLM-Malaria Dataset.

Overall, the submission is well organized, and the innovative parts are just fit. However, the mean average precision 75.9% is not good enough if compared with the cited references. Followings are some suggestions on the improvements.

1. The section number should begin from 1.
2. There are several typos and grammar errors which could easily be checked by some editing tools.
3. There is a big problem to use NWD because not all the classes are ellipse-like. This is why the mAP is only 75.9%. This could be verified from Table 8 where NWD+CIOU seems not so crucial for the detetcion. The authors should try some other loss functions.
4. The comparisons with the cited related works [11-18] should be provided to prove the effectiveness of the proposed methodology.

Reviewer 3 Report

Comments and Suggestions for Authors

The manuscript "MAS_Net: A Novel Malaria Cell Detection Algorithm" presents an approach for malaria detection in blood smears using a deep learning model. The key components of the model include the Split Contextual Attention Structure (SPCot) for enhanced feature extraction and the Multi-scale Receptive Field Detection Head (MRFH) for adapting to varying target sizes. The manuscript reports an average accuracy of 75.9% on the NLM-Malaria Dataset, demonstrating the model's competitiveness with current state-of-the-art methods.

The manuscript introduces a novel and potentially impactful method for malaria detection. Its strength lies in the innovative use of deep learning techniques tailored to the specific challenges of malaria cell detection. However, the following issues should be addressed:

1.       In the manuscript, the MAS-Net architecture, particularly with the integration of the Split Contextual Attention (SPCot) module and the Multi-scale Receptive Field Head (MRFH), introduces significant complexity. This complexity could potentially lead to high computational demands, which may limit its practical applicability, especially in resource-limited settings where such advanced computational resources may not be readily available. A more detailed analysis of the algorithm's computational efficiency could include:

·        * An assessment of the computational resources (like GPU memory, processing power) required to run MAS-Net effectively, which is crucial for understanding its feasibility in different clinical settings.

·         * Evaluation of the time taken to process an image or a batch of images.

·         * Suggestions for optimizing the algorithm to reduce computational load without significantly compromising performance. This could involve techniques like model pruning, quantization, or the use of more efficient network architectures.

·        * Discussion on how the model scales with increasing data volume or image resolution.

·        * Exploration of the trade-offs between model complexity (and thus, its diagnostic accuracy) and computational efficiency.

2.       In the manuscript, the introduction of novel metrics such as CIOU and NWD requires a detailed explanation to understand their specific relevance and benefits in the context of malaria cell detection:

·      * The manuscript should explain why these particular metrics are chosen for this application. For instance, CIOU might be particularly useful in assessing the accuracy of bounding box predictions in segmenting malaria parasites in blood smear images. NWD could be relevant in evaluating the distributional similarity between predicted and actual parasite locations.

·       * The manuscript needs to articulate the advantages these metrics offer over more traditional metrics such as precision, recall, or standard Intersection over Union (IoU). This might include how CIOU and NWD provide a more nuanced or accurate assessment of the model's performance in specific aspects relevant to malaria detection.

·       * Providing comparative analysis showing the effectiveness of these metrics in improving model performance compared to traditional metrics would strengthen the argument for their inclusion.

3.       A more comprehensive comparison with other state-of-the-art methods, including non-deep learning techniques, would provide a better understanding of MAS-Net's advantages and limitations. The comparison could involve multiple performance metrics such as accuracy, sensitivity, specificity, and F1 score. These metrics provide a multifaceted view of performance, highlighting strengths and weaknesses in different areas.

4.       Deep learning models, especially in healthcare, face challenges in interpretability. The manuscript could explore methods to make the model's decision-making process more transparent for clinicians:

·       * Implementing techniques to visualize what the network's layers are focusing on can provide insights. Techniques like Grad-CAM (Gradient-weighted Class Activation Mapping) can highlight the regions in the input images that are important for predictions, helping clinicians understand why the model makes certain decisions.

·       * Identifying and presenting the most critical features used by the model for decision-making can aid in interpretability. This could involve analyzing the weights and activations within the network to understand which features (e.g., cell size, shape, color) are most influential in malaria cell detection.

·       * Incorporating frameworks that provide explanations for model predictions, such as LIME (Local Interpretable Model-agnostic Explanations) or SHAP (SHapley Additive exPlanations), can make the model's decision-making process more transparent. These tools explain the output of the model in a more human-understandable manner.

5.       Conducting a robustness analysis against various types of noise and artifacts common in medical images would be valuable. Additionally, an error analysis to understand the types of cases where MAS-Net underperforms or fails could guide future improvements:

·       *Testing the model's performance in the presence of common distortions in medical images, such as variations in lighting, blurring, and artifacts from image capture, is crucial. This can involve artificially introducing these factors into the dataset or using datasets that naturally contain such variations. The goal is to assess how well the model maintains its accuracy under less-than-ideal conditions.

·       * Analyzing the types of errors made by MAS-Net is vital for understanding its limitations. This involves examining false positives and false negatives closely. For example, understanding if errors are more common in images with multiple cells, poor contrast, or overlapping features can provide insights into where the model needs improvement.

·      * Investigating MAS-Net's sensitivity to different stages of malaria is sigmificant because the Plasmodium parasite, responsible for malaria, undergoes several morphological changes during its lifecycle in human blood. These stages include ring, trophozoite, schizont, and gametocyte stages, each with distinct appearances. The model's ability to accurately identify and differentiate these stages is vital for effective diagnosis and treatment planning. If the model primarily detects malaria in one stage but not others, it might miss infections at different stages, leading to inaccurate diagnoses. Testing the model's performance across all stages ensures comprehensive detection capabilities, making it more reliable and effective in various clinical scenarios.

6.       The manuscript should explicitly state any limitations and propose directions for future research:

·       * Limitations might include the model's dependency on high-quality images, its performance variability across different Plasmodium species, or its applicability only to specific stages of the parasite's lifecycle.

·       * Potential directions could include:

o   Expanding the model to detect other blood-borne diseases in smear images, thus increasing its utility.

o   Enhancing the model's performance with more diverse datasets, including those from different geographical regions or with varying image quality.

o   Integrating the model into a larger diagnostic framework that includes patient history and other diagnostic tests.

o   Exploring real-time diagnosis capabilities for use in field settings.

Round 2

Reviewer 1 Report

Comments and Suggestions for Authors

The authors have given sufficient consideration to all points of criticism. Just as a final remark: In l.17 simply say "multi-scale detection head".

Reviewer 3 Report

Comments and Suggestions for Authors

The manuscript can be accepted in its present form.